# Immp2l Enhances the Structure and Function of Mitochondrial Gpd2 Dehydrogenase

**DOI:** 10.3390/ijms25020990

**Published:** 2024-01-12

**Authors:** Raymond A. Clarke, Hemna Govindaraju, Martina Beretta, Ellen Olzomer, Adam J. Lawther, Adam K. Walker, Zhiming Fang, Valsamma Eapen, Tzipi Cohen Hyams, Murray Killingsworth, Wallace Bridge, Nigel Turner, Khawar Sohail Siddiqui

**Affiliations:** 1Discipline of Psychiatry, University of New South Wales, Sydney, NSW 2052, Australia; a.walker@neura.edu.au (A.K.W.); v.eapen@unsw.edu.au (V.E.); 2Ingham Institute for Applied Medical Research, Sydney, NSW 2170, Australia; z.fang@unswalumni.com (Z.F.); tzipi.cohen-hyams@inghaminstitute.org.au (T.C.H.); murray.killingsworth@health.nsw.gov.au (M.K.); 3Academic Unit of Infant Child and Adolescent Services (AUCS), South Western Sydney Local Health District, Liverpool, NSW 2170, Australia; 4Department of Pharmacology, School of Biomedical Sciences, University of New South Wales, Sydney, NSW 2052, Australia; h.govindaraju@victorchang.edu.au (H.G.); n.turner@victorchang.edu.au (N.T.); 5Victor Chang Cardiac Research Institute, Darlinghurst, NSW 2010, Australia; 6School of Biotechnology and Biomolecular Sciences, University of New South Wales, Sydney, NSW 2052, Australia; m.beretta@unsw.edu.au (M.B.); e.olzomer@unsw.edu.au (E.O.); wj.bridge@unsw.edu.au (W.B.); 7Laboratory of ImmunoPsychiatry, Neuroscience Research Australia, Randwick, NSW 2031, Australia; adamlawther@gmail.com; 8Monash Institute of Pharmaceutical Sciences, Monash University, Parkville, VIC 3052, Australia; 9NSW Health Pathology, Liverpool Hospital Campus, Liverpool, NSW 2107, Australia

**Keywords:** enzyme structure, mitochondrial dynamics, mitochondrial size, NAD^+^ biosynthesis, organ mass, behaviour, autism, mitochondrial respiration, non-mitochondrial respiration, lean body mass, glutathione

## Abstract

‘Inner mitochondrial membrane peptidase 2 like’ (IMMP2L) is a nuclear-encoded mitochondrial peptidase that has been conserved through evolutionary history, as has its target enzyme, ‘mitochondrial glycerol phosphate dehydrogenase 2′ (GPD2). IMMP2L is known to cleave the mitochondrial transit peptide from GPD2 and another nuclear-encoded mitochondrial respiratory-related protein, cytochrome C1 (CYC1). However, it is not known whether IMMP2L peptidase activates or alters the activity or respiratory-related functions of GPD2 or CYC1. Previous investigations found compelling evidence of behavioural change in the *Immp2l^KD^*−/− KO mouse, and in this study, EchoMRI analysis found that the organs of the *Immp2l^KD^*−/− KO mouse were smaller and that the KO mouse had significantly less lean mass and overall body weight compared with wildtype littermates (*p* < 0.05). Moreover, all organs analysed from the *Immp2l^KD^*−/− KO had lower relative levels of mitochondrial reactive oxygen species (mitoROS). The kidneys of the *Immp2l^KD^*−/− KO mouse displayed the greatest decrease in mitoROS levels that were over 50% less compared with wildtype litter mates. Mitochondrial respiration was also lowest in the kidney of the *Immp2l^KD^*−/− KO mouse compared with other tissues when using succinate as the respiratory substrate, whereas respiration was similar to the wildtype when glutamate was used as the substrate. When glycerol-3-phosphate (G3P) was used as the substrate for Gpd2, we observed ~20% and ~7% respective decreases in respiration in female and male *Immp2l^KD^*−/− KO mice over time. Together, these findings indicate that the respiratory-related functions of mGpd2 and Cyc1 have been compromised to different degrees in different tissues and genders of the *Immp2l^KD^*−/− KO mouse. Structural analyses using AlphaFold2-Multimer further predicted that the interaction between Cyc1 and mitochondrial-encoded cytochrome b (Cyb) in Complex III had been altered, as had the homodimeric structure of the mGpd2 enzyme within the inner mitochondrial membrane of the *Immp2l^KD^*−/− KO mouse. mGpd2 functions as an integral component of the glycerol phosphate shuttle (GPS), which positively regulates both mitochondrial respiration and glycolysis. Interestingly, we found that nonmitochondrial respiration (NMR) was also dramatically lowered in the *Immp2l^KD^*−/− KO mouse. Primary mouse embryonic fibroblast (MEF) cell lines derived from the *Immp2l^KD^*−/− KO mouse displayed a ~27% decrease in total respiration, comprising a ~50% decrease in NMR and a ~12% decrease in total mitochondrial respiration, where the latter was consistent with the cumulative decreases in substrate-specific mediated mitochondrial respiration reported here. This study is the first to report the role of Immp2l in enhancing Gpd2 structure and function, mitochondrial respiration, nonmitochondrial respiration, organ size and homeostasis.

## 1. Introduction

IMMP2L and GPD2 have been conserved throughout evolutionary history from *E coli*, yeast, mice and humans [1,2]. IMMP2L is a nuclear-encoded mitochondrial peptidase that cleaves the signal transit peptides from other nuclear-encoded mitochondrial proteins including two respiratory-related proteins, cytochrome c1 (CYC1) and mitochondrial glycerol phosphate dehydrogenase 2 (GPD2) (Figure 1). CYC1 is an integral component of the mitochondrial respiratory (electron transport) chain (ETC), and GPD2 forms an integral component of the glycerol phosphate shuttle (GPS) that transfers electrons from the NADH generated in glycolysis to the mitochondrial ETC to mediate mitochondrial respiration, generate ATP and regenerate NAD^+^ for another round of glycolysis (Figure 1) [3]. However, prior to this study, it was not known whether IMMP2L had any effect on the function and respiratory-related roles of CYC1 or GPD2 within the mitochondria.

Prior investigations of a mouse model expressing a C-terminal truncated form of Immp2l returned conflicting results with respect to Immp2l’s role in cellular and mitochondrial respiration [4,5,6,7]. In 2011, Ma et al., reported a decrease in mitochondrial respiration in Immp2l^+/−^ heterozygotes carrying the C-terminal truncated form of Immp2l^+/−^, and in 2014, Bharadwaj et al., reported no decrease in mitochondrial respiration in an Immp2l^−/−^ homozygote with the truncated form of Immp2l^−/−^ [4,5,6,7]. These seemingly contradictory findings for Immp2l-mediated mitochondrial respiration possibly involved loss- and/or gain-of-function interference effects arising from Immp1l-Immp2l truncated heterodimers as reported by Lawther et al., in 2023 [3]. While other investigators had erroneously assumed that loss of Immp2L peptidase activity caused complete loss of Gpd2 enzyme activity and cellular senescence [8], infertility, neurodegeneration and early aging [4,5,6,7] in the present study we report that Gpd2 is not inactivated when Immp2l is knocked-out and we find no evidence of cellular senescence, infertility, neurodegeneration, oxidative stress or early aging in the *Immp2l^KD^*−/− KO mouse [3,9,10,11]. Notwithstanding, the present study represents the first respiratory analysis of Immp2l using a ‘clean’ Immp2l KO mouse model (*Immp2l^KD^*−/− KO mouse) without interference effects from truncated Immp2l proteins [3]. Furthermore, the *Immp2l^KD^*−/− KO mouse is devoid of all Immp2l peptidase activity, as confirmed by its failure to cleave and remove the signal transit peptides from Cyc1 and Gpd2 [3]. Moreover, in the present study we applied multiple respiratory substrates, separately, including Glycerol-3-Phosphate (G3P) with mitochondria isolated from a range of different tissues of the *Immp2l^KD^*−/− KO mouse, including brain, heart, kidney and brown adipose tissue.

Of great interest in the present study was the finding that all organs tested in the *Immp2l^KD^*−/− KO mouse were smaller compared with their wildtype litter mates. Moreover, this decrease in organ size was consistent with an earlier finding that all organs tested in the *Immp2l^KD^*−/− KO mouse had lower levels of mitochondrial ROS (mitoROS) with the greatest decrease in the kidney [3]. In the present study, the greatest decrease in mitochondrial respiration was also found in the kidney of the *Immp2l^KD^*−/− KO mouse when using succinate as the respiratory substrate. These decreases in mitochondrial respiration and mitoROS were consistent with deficits in the mitochondrial respiratory chain. In this study, we identified structural changes in the uncleaved Cyc1 and Gpd2 proteins of the *Immp2l^KD^*−/− KO mouse, which affect their interactions with adjacent proteins. Firstly, we identified variant molecular connections between the ‘uncleaved’ Cyc1 and the Cyb respiratory subunit of Complex III in the *Immp2l^KD^*−/− KO mouse. Secondly, when using G3P as the substrate for Gpd2, we observed further decreases in mitochondrial respiration of ~20% and 7%, respectively, in female and male *Immp2l^KD^*−/− KO mice. These decreases were associated with structural changes in the homodimerization of the Gpd2 dehydrogenase.

## 2. Results

### 2.1. Immp2l^KD^−/− KO Is Associated with Lower Lean Body Mass

Male and female *Immp2l^KD^*−/− KO mice have decreased body weight compared with wildtype mice. Male *Immp2l^KD^*−/− KO mice displayed decreased body mass compared with wildtype mice at 22–29 weeks of age (t(30) = 2.234, *p* = 0.033; Figure 2A). Using EchoMRI, we observed no difference in fat mass between the *Immp2l^KD^*−/− KO mouse and wildtype litter mates (Figure 2B), but rather lower lean body mass (Figure 2C). At the time of sacrifice at 25–32 weeks of age, body mass in *Immp2l^KD^*−/− KO mice was decreased compared to their wildtype littermates (t(30) = 2.234, *p* = 0.033; Figure 2), with lower mass observed in all organs tested from the *Immp2l^KD^*−/− KO mice (Figure 2E–L), reaching significance in heart, quadriceps and kidney (Figure 2E–G). When normalised to body mass, there were no significant differences in fat mass, lean mass, organ weights or adipose tissue depots, indicating that the lower body mass of *Immp2l^KD^*−/− KO mice is likely due to the proportionally lower mass of the organs, rather than a decrease in size of any specific organ or a disproportionate decrease in fat mass.

### 2.2. Oxygen Consumption Rates in Mitochondria Isolated from Heart, Brain and Kidney

Respiration rates were determined for mitochondria isolated from the kidney, heart and brain using a Clark-type oxygen electrode and different substrates including glutamate that feeds electrons into Complex I and succinate which feeds electrons into Complex II (Figure 1). When using glutamate as the substrate, we detected no significant difference in oxygen consumption between the mitochondria isolated from the *Immp2l^KD^*−/− KO mouse and their wildtype litter mates (Figure 3). When using succinate as the substrate, we detected a decrease in oxygen consumption in mitochondria isolated from the kidney of the *Immp2l^KD^*−/− KO mouse when compared with wildtype litter mates but not in mitochondria from heart or brain (Figure 3). Oxygen consumption was significantly lower in kidney mitochondria from the *Immp2l^KD^*−/− KO mouse compared to wildtype litter mates (*p* = 0.03) under non-phosphorylating state 2 respiration, when there is substrate present, but no ADP. Under state 3 respiration conditions (ADP present) and state 4 (ATP synthesis inhibited by oligomycin), respiration was still trending lower in the *Immp2l^KD^*−/− KO mice, although the differences did not reach statistical significance. Overall, there was an indication that mitochondrial respiration with succinate may be lower in the *Immp2l^KD^*−/− KO kidney but not the other organs tested.

### 2.3. Oxygen Consumption Rates in Mitochondria Isolated from Mouse BAT Tissue

Seahorse XF96 analysis was used to test oxygen consumption rates in mitochondria isolated from brain and brown adipose tissue. Glycerol-3-phosphate (G3P) was used as the substrate for the Gpd2 enzyme. Gpd2 dependent mitochondrial respiration was too low in the brain of wildtype mice for comparative analysis. Given that Gpd2 is expressed at much higher levels in brown adipose tissue (BAT) [12] we isolated BAT mitochondria from mice and used G3P as the substrate for Gpd2 to determine differences in G3P/Gpd2 dependent mitochondrial respiration (Figure 4).

Seahorse analysis of oxygen consumption rates (OCR) in mouse BAT tissue mitochondria indicated that female and male *Immp2l^KD^*−/− KO mice had a ~20% and ~7% decrease, respectively, in G3P/Gpd2 enzyme-dependent mitochondrial respiration in *Immp2l^KD^*−/− KO mice compared with the wildtype (Figure 4). Notwithstanding, G3P/Gpd2-dependent mitochondrial respiration was over twice as high in females compared with males (Figure 4).

### 2.4. Predictive Structural Analysis of Mouse mGpd2 Variant Homodimers using AlphaFold2-Multimer

The lower Gpd2-dependent respiratory levels in mitochondria isolated from the *Immp2l^KD^*−/− KO mouse suggested that there was a decrease in the activity of the ‘uncleaved Gpd2′ enzyme and/or interference of its role within the glycerol phosphate shuttle. We used AlphaFold2-Multimer predictive alignment error (PAE) plots visualised through ChimeraX [13] to interrogate the homodimeric structure and associated inter-subunit connections between the mouse Gpd2 dehydrogenase homodimer variants. AlphFold2-Multimer analysis [13,14,15] predicted with high confidence that the ‘cleaved Gpd2′ dehydrogenase in wildtype mice and the ‘uncleaved Gpd2′ enzyme in *Immp2l^KD^*−/− KO mice both form homodimers, albeit the latter ‘uncleaved mGpd2′ homodimer in mouse forms 6% more inter-subunit connections (Figure 5). This indicated a fundamental change in the structure of the Gpd2 dehydrogenase homodimer within the inner mitochondrial membrane where it is cleaved and where it functions within the glycerol phosphate shuttle (GPS). The GPS shuttle facilitates the transfer of electrons from NADH (from glycolysis) to FAD (embedded within Gpd2) and onto Complex II of the mitochondrial respiratory chain (Figure 1). The AlphaFold2-Multimer analysis also predicted that the signal transit peptides of the ‘uncleaved Gpd2′ enzyme homodimer protrude further into the surrounding inner mitochondrial membrane from where the Gpd2 dehydrogenase is normally embedded (Figure 6) and from where electrons are transferred between the FADH_2_ generated by Gpd2 in the GPS and Complex II of the electron transport chain.

### 2.5. Predictive Structural Analysis of Mouse Cyc1 Variants in Complex with Cyb Using AlphaFold2-Multimer

Three respiratory subunits in Complex III of the electron transport chain interact, namely Cyc1, Cyb and UQCRFS1. AphaFold2-Multimer was used to predict the orientation of these three subunits relative to each other. The orientation of Cyc1 with respect to the other two subunits is such that the ‘uncleaved’ signal transit peptide of Cyc1 (as found in the *Immp2l^KD^*−/− KO mouse) was observed to align in closer proximity to Cyb than to UQCRFS1. To investigate this further, it was queried whether the ‘uncleaved’ transit peptide of Cyc1 established any ‘new’ intermolecular connections with Cyb. Alpha-Fold2-Multimer predicted 56 ‘new’ molecular contacts between the signal transit peptide (Orange) of Cyc1 (Blue chain) and Cyb (Pink chain) (Figure 7). These contacts involved the following residues: Transit Peptide of Cyc1: Met4, Leu8, Leu11, Ala19, Lue22, His23, Val26. Normal Cyb: Gln322, Trp326, Ile327, Trp337, Gln341, Pro346, Phe347, Ile350 (Figure 7D).

### 2.6. Immp2l KO Mediated Decrease in Mitochondrial and Nonmitochondrial Respiration

Seahorse analysis was used to test for differences in respiration rates between living primary mouse embryonic fibroblast cell lines (MEFs) derived from *Immp2l^KD^*−/− KO and wildtype day 15 embryos. Primary MEFs were grown in DMEM and harvested during the growth phase at ~80% confluence. The total level of cellular respiration was ~27% lower in MEFs derived from *Immp2l^KD^*−/− KO mice (H3 and H10) compared with wildtype (WT) (Figure 8, Table 1). This deficit in respiration in *Immp2l^KD^*−/− KO MEFs comprised a dramatic ~50% drop in nonmitochondrial respiration and a modest ~12% decrease in mitochondrial respiration (Figure 8A, Table 1). This observation was somewhat surprising given that Immp2l is a mitochondrial peptidase yet nonmitochondrial respiration was more greatly affected than mitochondrial respiration in MEFs derived from *Immp2l^KD^*−/− KO mice.

## 3. Discussion

Our previous investigations found compelling evidence of behavioural change in the *Immp2l^KD^*−/− KO mouse that was more pronounced in male compared with female mice [3,16,17]. In this study, male and female *Immp2l^KD^*−/− KO mice had lower body mass compared with wildtype litter mates. Using EchoMRI, male *Immp2l^KD^*−/− KO mice were shown to have a lower lean body mass compared with wildtype litter mates, while fat mass was unaffected by genotype. What was most interesting in this analysis was that all organs examined were observed to be smaller in size in the *Immp2l^KD^*−/− KO mouse compared with wildtype litter mates, with the drop in total lean body mass appearing to represent the cumulative sum of the lower masses of the different organs. Furthermore, we found no marked difference in the mass of two separate fat pads, the epididymal fat pad (EpiWAT) and the inguinal fat pad (IngWAT)) of the *Immp2l^KD^*−/− KO mouse compared with wildtype litter mates (Figure 2). This finding was in contrast with the lower fat mass reported for a C-terminal truncated *Immp2l* mouse model [3,4] and the lower weight loss reported for a Gpd2 KO mouse line [18]. These results identify a key role for Immp2l in regulating the homeostasis of the organs of the body.

In an earlier study, we demonstrated that a loss of Immp2l peptidase activity in the *Immp2l^KD^*−/− KO mouse resulted in the failure of Immp2l to cleave and remove the mitochondrial transit peptides from two of its nuclear-encoded respiratory-associated mitochondrial target proteins, Cyc1 and Gpd2 [3]. Cyc1 forms part of Complex III of the mitochondrial respiratory chain, and the Gpd2 enzyme forms an integral component of the glycerol phosphate shuttle (GPS), which funnels electrons from the NADH generated during glycolysis in the cytosol to the mitochondrial respiratory chain and in doing so regenerates NAD^+^ for another round of glycolysis (Figure 1). This loss of Immp2l peptidase activity was also associated with a decrease in the level of mitochondrial ROS (mitoROS) to different degrees in different tissues [3]. However, prior to the present study, it was not known whether Immp2l had any effect on cellular or mitochondrial respiration rates or the activity or function of Cyc1 or Gpd2 [3,4,5,6,7].

In the present study, we used the *Immp2l^KD^*−/− KO mouse model to investigate whether Immp2l has a role in the regulation of respiration. In this endeavour, we used a comprehensive approach by testing for differences in respiration between mitochondria isolated from different organs and tissues (kidney, heart, brain and brown adipose tissue) using different respiratory substrates (glutamate, succinate and glycerol-3-phosphate (G3P)). Notwithstanding, all respiratory pathways tested feed electrons into the mitochondrial electron transport chain (ETC) upstream of Cyc1/Complex III. Glutamate generates NADH that feeds electrons directly into Complex I, and succinate feeds electrons directly into Complex II of the ETC, whereas glycerol-3-phosphate (G3P) is a cytosolic substrate for Gpd2 within the GPS. G3P is converted by Gpd2 into dihydroxyacetone phosphate (DHAP) in the GPS which mediates the transfer of electrons from the cytosolic NADH produced during glycolysis to FAD^2+^ from where the electrons are transferred through the inner mitochondrial membrane to Complex II of the mitochondrial respiratory chain (see Figure 1). We also tested for differences in whole cell respiration using living fibroblast cell lines derived from the *Immp2l^KD^*−/− KO mouse.

When using glutamate as the respiratory substrate, we found no difference in the level of mitochondrial respiration in brain, heart or kidney of the *Immp2l^KD^*−/− KO mouse compared with wildtype littermates. When using succinate as the respiratory substrate we observed a ~31% lower level of mitochondrial respiration in the kidney but not in brain or heart of the *Immp2l^KD^*−/− KO mouse. This result was consistent with the ~50% decrease in mitoROS found in the kidney of the *Immp2l^KD^*−/− KO mouse [3] and the much lower decreases in mitoROS observed in brain and heart. The differences in the level of mitochondrial respiration between tissues and substrates are difficult to explain without further testing; notwithstanding, Complex III is composed of numerous different subunits, which may differ proportionally under different conditions. An earlier 2011 study reported a decrease in mitochondrial respiration in the brain of Immp2l^+/−^ heterozygous mice expressing a C-terminal truncated form of Immp2l when combining glutamate, malate and succinate as the respiratory substrate, whereas a 2014 study reported no decrease in mitochondrial respiration in skeletal muscle from an Immp2l^−/−^ homozygous mouse expressing the same C-terminal truncated form of Immp2l when using succinate alone as the substrate [4,5,6,7]. Notwithstanding, these earlier findings may have been confounded by loss- and/or gain-of-function effects involving Immp1l–Immp2l truncated heterodimers as described by Lawther et al., in 2023 [3]. Moreover, these prior investigations of mice expressing the C-terminal truncated form of Immp2l did not test G3P as a respiratory substrate, thus limiting the ability to make comparisons between studies.

To investigate the molecular basis of the decreased mitochondrial respiration in our clean *Immp2l^KD^*−/− KO mice, we analysed the structure of the uncleaved Cyc1 found in the *Immp2l^KD^*−/− KO mouse and its molecular connections with another closely associated respiratory subunit Cyb (Figure 6). This analysis predicted with confidence that the failure to remove the signal transit peptide from Cyc1 in the *Immp2l^KD^*−/− KO mouse had altered molecular connections between the Cyc1 and Cyb respiratory subunits of Complex III. In this analysis, Alpha-Fold2-Multimer predicted 56 ‘new’ molecular interactions between the ‘uncleaved’ signal transit peptide of Cyc1 and Cyb in mice (Figure 7D). How these ‘new’ molecular interactions between the two respiratory subunits might affect respiration is uncertain. However, together these results indicated a deficit in mitochondrial respiration in the kidney of the *Immp2l^KD^*−/− KO mouse due to a partial/conditional deficit in the function of Cyc1-Cyb in Complex III; however, this interpretation of the results requires further respiratory studies using additional tissues.

We used G3P as the substrate for Gpd2 in mitochondria isolated from brown adipose tissue (BAT). Gpd2 is known to be expressed at high levels in BAT compared with other tissues [9]. Using this approach, we identified a ~20% and ~7% lower level of Gpd2-dependent mitochondrial respiration over time in female and male *Immp2l^KD^*−/− KO mice, respectively, compared with wildtype. Results indicated there was over a 2-fold higher level of Gpd2-dependent mitochondrial respiration in females compared with males. As a result, the levels of Gpd2-dependent mitochondrial respiration were always much higher in female mice compared with male mice regardless of genotype. This finding was most interesting given that hemizygous deletions of *GPD2* are more frequent in male compared with female autism, and a deleterious hemizygous mutation near the active site of GPD2 has been found segregating with male autism in a multi-generation autism family [19,20]. Secondly, autism more generally occurs with a much higher incidence in males compared with females [20]. Thirdly, behavioural change is more pronounced in male compared with female *Immp2l^KD^*−/− KO mice [3]. Given these GPD2 associations with autism/behavioural change, it is possible that the lower comparative incidence of autism in females with hemizygous deletions of GPD2 may be due to a much higher expression level of GPD2 in females.

The above finding further indicated that failure to remove the signal transit peptide from Gpd2 by Immp2l had compromised the activity of the Gpd2 dehydrogenase and/or the role of Gpd2 in the GPS, wherein electrons are transferred from NADH to FAD^2+^ and onto Complex II. To determine if there was any molecular basis for this deficiency in Gpd2 function, we searched for structural changes in Gpd2 using AlphaFold2-Multimer predictive structural analysis and found that failure to remove the transit peptide from Gpd2 in the *Immp2l^KD^*−/− KO mouse had altered the structure of the Gpd2 enzyme in a number of ways that could affect respiration. Firstly, this structural change altered the homodimerization of Gpd2 by increasing the number of inter-subunit connections in the Gpd2 homodimer. Secondly, the transit peptides that remained attached to the Gpd2 homodimer in the *Immp2l^KD^*−/− KO mouse extended further into the surrounding inner mitochondrial membrane, where they could feasibly interfere with the Gpd2/FADH2-mediated transfer of electrons to Complex II of the ETC.

To better understand the effect of Immp2l on whole cell respiration, we generated primary fibroblast cell lines (MEFs) from *Immp2l^KD^*−/− KO mouse embryos. The total level of respiration was decreased ~27% in primary MEFs derived from *Immp2l^KD^*−/− KO mice when compared with wildtype MEFs (Figure 8, Table 1). This ~27% deficit in total cellular respiration in *Immp2l^KD^*−/− KO MEFs comprised a dramatic ~50% decrease in nonmitochondrial respiration concurrent with a more modest ~12% decrease in total mitochondrial respiration (Figure 8B, Table 1). This ~12% decrease in total mitochondrial respiration in *Immp2l^KD^*−/− KO MEFs was consistent with the cumulative decreases we had observed in substrate-specific mediated mitochondrial respiration in tissues from the *Immp2l^KD^*−/− KO mouse (Figure 3 and Figure 4). Notwithstanding, evidence suggests that substrate specific mitochondrial respiration rates vary widely between substrates and tissues. However, the source and impact of the ~50% decrease in non-mitochondrial respiration/oxygen consumption remain unexplained.

Together, these results identify key roles for Immp2l in enhancing and maximising mitochondrial respiration, glycolysis and nonmitochondrial respiration (NMR). The Immp2l peptidase enhances mitochondrial respiration by cleaving and thereby enhancing the structure and function of two respiratory-related mitochondrial proteins Gpd2 and Cyc1. In this respect, the enhanced structure of the Gpd2 dehydrogenase homodimer induced by the Immp2l peptidase was of particular interest given that both Immp2l and Gpd2 have been conserved through evolutionary history from *E coli* to human. We report that the cleavage of Gpd2 by Immp2l increased the number of molecular connections between the subunits of the Gpd2 homodimer; notwithstanding, the precise effect of this change on homodimerization requires further research. Furthermore, it is possible that this decrease in mitochondrial respiration is related to the reduction in ROS levels and/or the reduction in organ and body size in these Immp2l KO mice, albeit further research is required to confirm this and to determine whether changes in metabolism and/or mitochondrial dynamics are also implicated. The decrease in the size of organs in the *Immp2l^KD^*−/− KO mice suggests that these decreases may be proportional and that the decrease in body size should not be mistaken for weight loss as fat mass remained unaffected in the *Immp2l^KD^*−/− KO mouse. Nonmitochondrial respiration could likewise be implicated in these physiological changes in the *Immp2l^KD^*−/− KO mouse. On the other hand, the behavioural changes in the *Immp2l^KD^*−/− KO mouse more likely involved changes in gene expression, available at https://genotypepress.com, accessed on 21 December 2023. In this respect, *GPD2* has been deleted and mutated in autism, and *IMMP2L* is strongly linked with autism inheritance; however, the molecular basis of these associations has yet to be established.

This study returned a number of unexpected results that limit our interpretation of the findings and which require further investigation. It is not certain why the level of substrate-mediated mitochondrial respiration differs between tissues. In this respect, the expression level of many of the subunits of complex III may vary between tissues and under different conditions; however, further investigations are required to understand this and the full impact of Immp2l on mitochondrial and cellular homeostasis. This line of investigation will require expansion of the number of MEFs derived from the *Immp2l^KD^*−/− KO mouse so as to better understand the full range and effect that Immp2l has on mitochondrial and nonmitochondrial respiration under different conditions. In this study, we did not quantify the impact of the Immp2l −/− KO on glycolysis or characterize the nature of the disproportionately large decrease in nonmitochondrial respiration, all of which are the subject of our ongoing studies into the role of Immp2l and its effects on mitochondrial dynamics, glycolysis and metabolism, in particular, its effect on NAD^+^ synthesis and gene regulation (Figure 1). In this respect, the *Immp2l^KD^*−/− KO mouse and derived MEFs provide an excellent model system to better understand the role of Immp2l in cellular homeostasis, mitochondrial respiration, glycolysis, metabolism, mitoROS production, organ size and behaviour and how this may affect the reprograming of cellular metabolism and its impact on NAD^+^ synthesis, gene expression and mitochondrial dynamics, available at https://genotypepress.com, accessed on 21 December 2023.

## 4. Materials and Methods

### 4.1. Animals and Ethics

Mice were bred and group housed at the Ingham Institute Biological Resource Unit (Liverpool, NSW, Australia) in individually ventilated cages (GM500 Green, Techniplast Australia Pty Ltd., Rydalmere, NSW, Australia) with corn cob bedding, crinkle-cut cardboard nesting material and a red igloo (Bioserv, Frenchtown, NJ, USA). At 6 months of age, mice were transported to the University of New South Wales (Sydney, NSW, Australia) for cellular respiration analyses and body composition, mitochondrial and gene expression analyses. Mice were allowed to acclimate to the new facility for one month before testing and were housed in standard holding cages in a temperature- and humidity-controlled environment with a 12/12 h modified dark–light cycle (lights on at 0700). Two mice were housed individually to avoid fighting. Food and water or MitoQ-treated water were available ad libitum. Mice were randomly allocated to experimental groups. Mice were euthanized with CO_2_ or cervical dislocation. Tissues were dissected after perfusion with sterile PBS. All procedures involving mice were carried out under protocols approved by the University of New South Wales Animal Ethics Committee (protocol number 19/6B, 15/48B and 18/78A) and in accordance with National Health and Medical Research Council guidelines. Animals were monitored daily. No unexplained mortality occurred in these studies.

### 4.2. Mouse Maintenance

Male *Immp2l^KD^*−/− KO mice and wildtype litter mates were obtained from the Ingham Institute Biological Resource Unit (Sydney, NSW, Australia). Mice were housed at 22 ± 1 °C in a 12:12 h light/dark cycle. Mice had ad libitum access to water and a standard low-fat rodent diet (Gordon’s Specialty Stock Feeds, Yanderra, NSW, Australia). Physiological assessments were approved by the UNSW animal care and ethics committee (ACEC 18/78A) and followed guidelines issued by the National Health and Medical Research Council of Australia.

### 4.3. Tissue Preparation and Mitochondrial Isolation

Frozen tissues were thawed from −80 °C in ice-cold PBS, minced and homogenized in MAS buffer (70 mM sucrose, 220 mM mannitol, 5 mM KH_2_PO_4_, 5 mM MgCl2, 1 mM EGTA, 2 mM HEPES pH 7.4). Heart and WAT were mechanically homogenized with 20 strokes in a glass Dounce homogeniser. Liver, BAT, brain, kidney, lung and muscle were mechanically homogenized with 10–20 strokes (depending on the tissue: lung and muscle require larger number of strokes) in a Teflon–glass homogeniser. All homogenates were centrifuged at 1000× *g* for 10 min at 4 °C; then, the supernatant was collected. Protein concentration was determined by the BCA method (ThermoFisher, VIC, Australia). The isolation of mitochondria followed methods previously described [21,22,23]. Immediately after excision, kidney and brain tissues were prepared by finely dicing in ice-cold isolation buffer A (250 mM sucrose, 10 mM Tris-HCl, 1 mM EGTA, 1% fatty-acid free BSA, pH 7.4). The tissue was rinsed with buffer A before homogenisation using a glass–Teflon Dounce homogeniser. Heart tissue was similarly prepared in ice-cold isolation buffer B (100 mM sucrose, 100 mM KCl, 50 mM Tris-HCl, 1 mM KH_2_PO_4_, 1 mM EGTA, 0.2% fatty-acid free BSA, pH 7.0). The heart tissue was incubated for 2 min in 1 mg/mL of proteinase and rapidly rinsed in buffer B to stop digestion. The heart tissue was then homogenised using a Polytron homogeniser (ThomasScientific, Swedesboro, NJ, USA). Homogenates were centrifuged at 1000× *g* for 5 min at 4 °C. The supernatant was collected and centrifuged at 10,000× *g* for 10 min at 4 °C. Pellets were washed in the respective isolation buffers and centrifuged again at 10,000× *g* for 10 min at 4 °C. The pelleted mitochondrial isolate was resuspended in BSA-free ice-cold isolation buffer A and assayed for protein quantification using the bicinchoninic acid (BCA) assay (ThermoFisher Scientific, Rockford, IL, USA).

### 4.4. Mitochondrial Respiration

Kidney, brain and heart mitochondrial isolates were resuspended in respiration buffer (225 mM mannitol, 75 mM sucrose, 10 mM Tris-HCl, 10 mM KH_2_PO_4_, 10 mM KCl, 0.1 mM EDTA, 0.8 mM MgCl_2_, 0.3% fatty-acid free BSA, pH 7.0) warmed to 37 °C. A Clark-type oxygen electrode (Rank Brothers, Cambridge, UK) was used to measure the oxygen consumption of mitochondrial isolates when provided with either succinate (5 mM) + rotenone (2 μM) or glutamate (5 mM) + malate (2 mM). Oxygen consumption of mitochondrial isolates was measured in the presence of the substrates alone (State 2), in the presence of 200 μM ADP (State 3) or in the presence of 2.5 μM oligomycin.

### 4.5. Mouse BAT Tissue Mitochondria Preparation and Respirometry

Brown adipose tissue (BAT) was processed using a modified protocol by Acin-Perez et al. [24]. Briefly, frozen samples were thawed from −80 °C in ice-cold PBS, minced and homogenized in 1 mL of MAS buffer (70 mM sucrose, 220 mM mannitol, 10 mM KH_2_PO_4_, 5 mM MgCl_2_, 1 mM EGTA, 2 mM HEPES; pH 7.2 with KOH). BAT tissues were mechanically homogenized using a Potter-Elvehjem homogeniser (8–10 strokes) and homogenates were centrifuged twice at 1000× *g* for 10 min at 4 °C. Then, supernatants were centrifuged at 10,000× *g* for 10 min at 4 °C, and mitochondrial pellets were washed twice in wash buffer (210 mM mannitol, 70 mM sucrose, 5 mM HEPES, 1 mM EGTA, 0.5% BSA; pH 7.2 with KOH). The final mitochondrial pellet was re-suspended in a small volume of ice-cold MAS buffer (75 µL starting from 30–60 mg of wet tissue) with no BSA and quantified (BCA, ThermoFisher). BSA was omitted from the final buffer to prevent interference with the protein assay kit and with the respirometry assay. Mitochondria were loaded 3 mg/well into Seahorse XF96 microplate in 50 mL of MAS. The loaded plate was centrifuged at 2000× *g* for 20 min at 4 °C (no brake), and an additional 130 microlitres of warm MAS (37 °C) was added to each well. To avoid disrupting mitochondrial adherence to the bottom of the plate, MAS was added using a multichannel pipette pointed at a 45° angle to the top of the well chamber. Mitochondria were equilibrated for 5 min at 37 °C without CO_2_ before starting the respirometry assay. Glycerol-3-phosphate (G3P, #94124 Sigma, Burlington, MA, USA) was injected at port A (5 mM final concentration, in MAS), and Antimycin A (A8674, Sigma) was injected at port B (4 mM final concentration, in MAS). Mix and measure times were 0.5 and 4 min, respectively. Oxygen consumption rate (OCR) was measured using an Agilent Seahorse XFe96 Analyzer (Agilent Technologies, Santa Clara, CA, USA), and Wave software (https://www.waveapps.com/, accessed on 21 December 2023) was used to export OCR rates to GraphPad Prism v.9.0.1. OCR rates were normalized to background noise measured in wells with no mitochondria. Note: MAS buffer is diluted 3× with water.

### 4.6. Mouse Embryonic Fibroblast Cell Lines (MEFs)

In this study, we crossed heterozygous *Immp2l^KD^* −/+ KO mice, and 14.5 days after fertilisation pregnant females were humanely euthanised. Then, under sterile conditions, individual mouse embryos were removed from the uterus one by one and transferred to individual sterile dishes containing sterile PBS. After the head and all red organs were removed, the remaining tissue was then transferred to individual 1.5 mL tubes containing 0.5 mL PBS. Tissue was minced with scissors before adding 0.5 mL digestion solution into each 0.5 mL of PBS in 1.5 mL Eppenorf tubes. Tubes were incubated at 37 °C for 1 h, shaking every 10 min. After 1 h, the mixture was removed from the Eppendorf tube and transferred to a 15 mL Falcon tube containing 9 mL of DMEM media and centrifuged at 1000 rpm for 3 min. Supernatant was discarded, and the cell pellet was resuspended in 10 mL of DMEM (with 10% FBS and Penicillin Streptomycin antibiotics) and transferred to a 10 cm cell culture dish. Cells were incubated and passaged regularly over an 8-week period until cultures were established.

### 4.7. Mouse Embryonic Fibroblast Cell Line (MEF) Preparation and Seahorse Analysis

MEF cell lines were trypsinised and centrifuged for 5 min at 1000 rpm. Real-time measurements of MEF oxygen consumption rate (OCR) were made using a Seahorse XF-96 analyser following the manufacturer’s instructions (Seahorse Biosciences, North Billerica, MA, USA). MEF cells (H3, WT7 and H10) were seeded into Seahorse 96-well tissue culture plates at a density of 20,000 cells per well and left for 24 h to adhere to the plate. Standard DMEM growth media was removed and replaced with Seahorse media (DMEM + 1 mM Sodium Pyruvate, 4 mM L-glutamine and 25 mM glucose). Cells were incubated at 37 °C in a non-CO2 incubator for 1 h prior to running the assay. Measurements were recorded with a 30 s mix and a 4 min measurement period [25]. Baseline measurements were followed by the addition of G3P (5 mM) and Antimycin A (4 µM). BCA assay was performed, and readout was normalised to protein content.

### 4.8. Body Composition Analysis

Whole-body composition data were assessed in 22–29-week-old mice, yielding lean mass, fat mass, free water mass and total water analysed by the EchoMRI-900 (EchoMRI Corporation Pty Ltd., Singapore) in accordance with the manufacturer’s instructions. Organ and fat weight were quantified in the same mice following sacrifice at 25–32 weeks of age, yielding mass for whole body, brown adipose tissue (BAT), inguinal white adipose tissue (ingWAT), epididymal white adipose tissue (epiWAT) brain, kidney, liver, heart and quadriceps. Whole-body composition data yielding lean mass, fat mass, free water mass, and total water were analysed by the EchoMRI-900 (EchoMRI Corporation Pty Ltd., Singapore) in accordance with the manufacturer’s instructions.

### 4.9. Respiratory and Body Composition Statistical Analyses

Mitochondrial respiration data were analysed using unpaired two-tailed *t*-tests to detect effects of genotype (*Immp2l^KD^* −/− KO vs. WT). Body composition data were analysed using unpaired two tailed *t*-tests to detect effects of genotype (Immp2l−/− vs. WT).

### 4.10. AlphaFold2-Multimer High-Accuracy Prediction of Complex Structures

High-accuracy predictions of protein structures and associated oligomerization states were carried out using the AlphaFold2-Multimer program in ChimeraX (v 1.6.1) available at https://www.rbvi.ucsf.edu/chimerax, accessed on 1 September 2023 [10] using default settings. Analysis of intermolecular contacts and visualization of structures were performed in ChimeraX [12] with predicted aligned error (PAE) plots interpreted as described [11].

## 5. Conclusions

Immp2l enhances the structure and function of Gpd2 and Cyc1, thereby increasing mitochondrial respiration. Immp2l also dramatically enhances the level of nonmitochondrial respiration by an as-yet-unidentified mechanism. As a positive regulator of Gpd2 function, we can assume that Immp2l also positively regulates glycolysis. Interestingly, Immp2l increases organ/body size independent of fat mass and further findings indicate that Immp2l also regulates mitochondrial size and NAD^+^ synthesis and gene expression, available at https://genotypepress.com, accessed on 21 December 2023. The *Immp2l^KD^*−/− KO mouse model and derived MEF cell lines represent an invaluable model resource to aid ongoing investigations into Immp2l’s impact on mitochondrial dynamics, NAD^+^ synthesis, gene expression, dopamine regulation and behaviour, available at https://genotypepress.com, accessed on 21 December 2023.

## Figures and Tables

**Figure 1 ijms-25-00990-f001:**
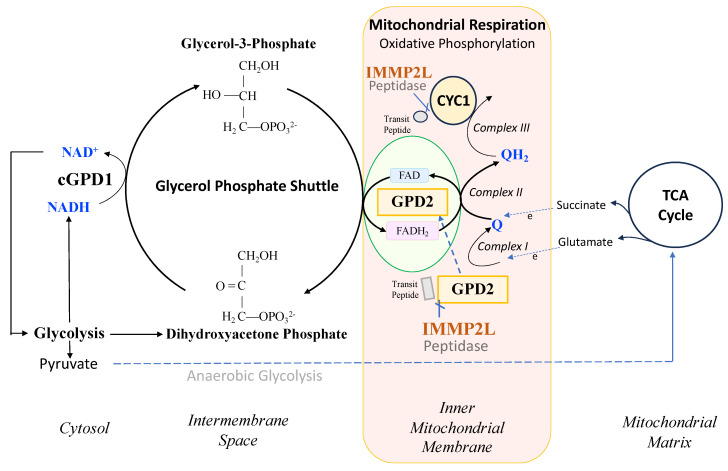
IMMP2L peptidase association with the glycerol phosphate shuttle that helps maximise mitochondrial respiration and glycolysis.

**Figure 2 ijms-25-00990-f002:**
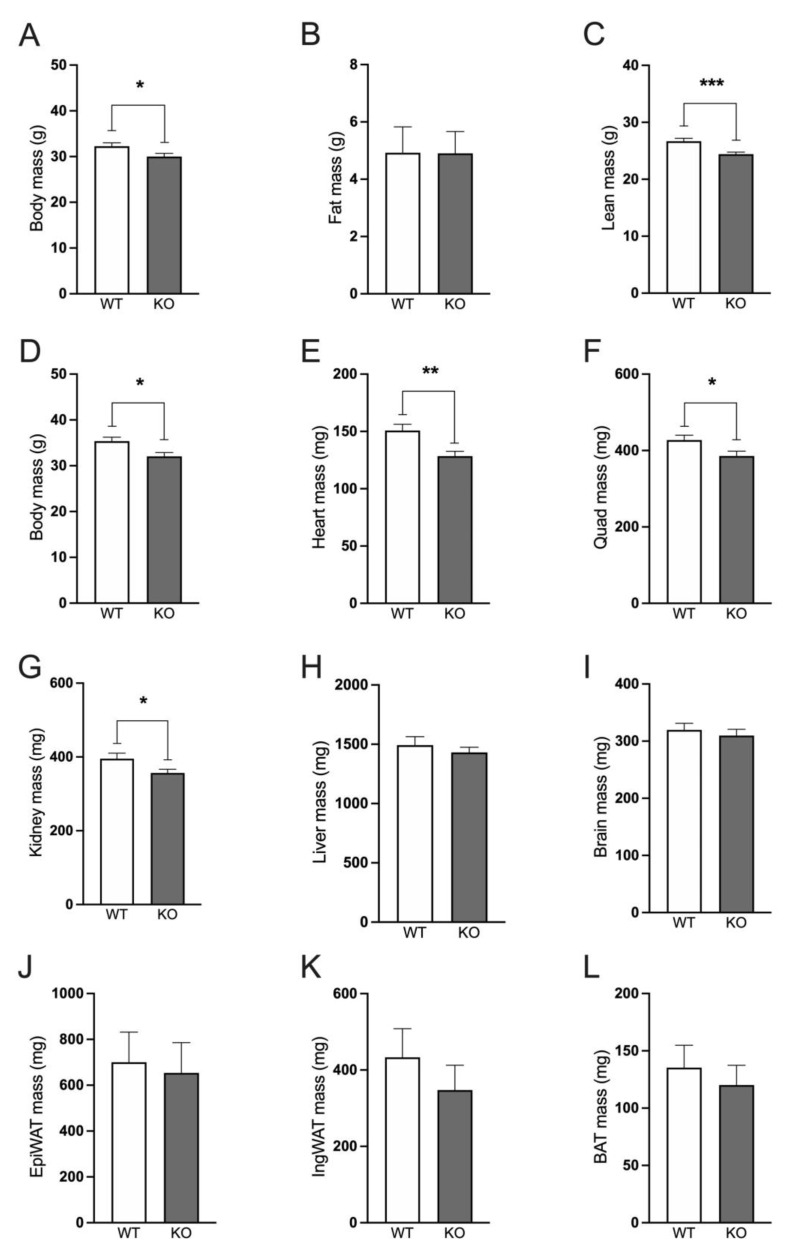
*Immp2l^KD^*−/− KO decreases lean body mass in mice. (**A**–**C**). Body mass, fat mass and lean mass of *Immp2l^KD^*−/− KO and wild-type mice at 22–29 weeks of age. (**D**–**L**). Body mass and organ weights of *Immp2l^KD^*−/− KO and wild-type mice at 25–32 weeks of age. Data represent mean ± SE (n = 7–16) and were analysed using an unpaired *t*-test * *p* < 0.05, ** *p* < 0.01, *** *p* < 0.001.

**Figure 3 ijms-25-00990-f003:**
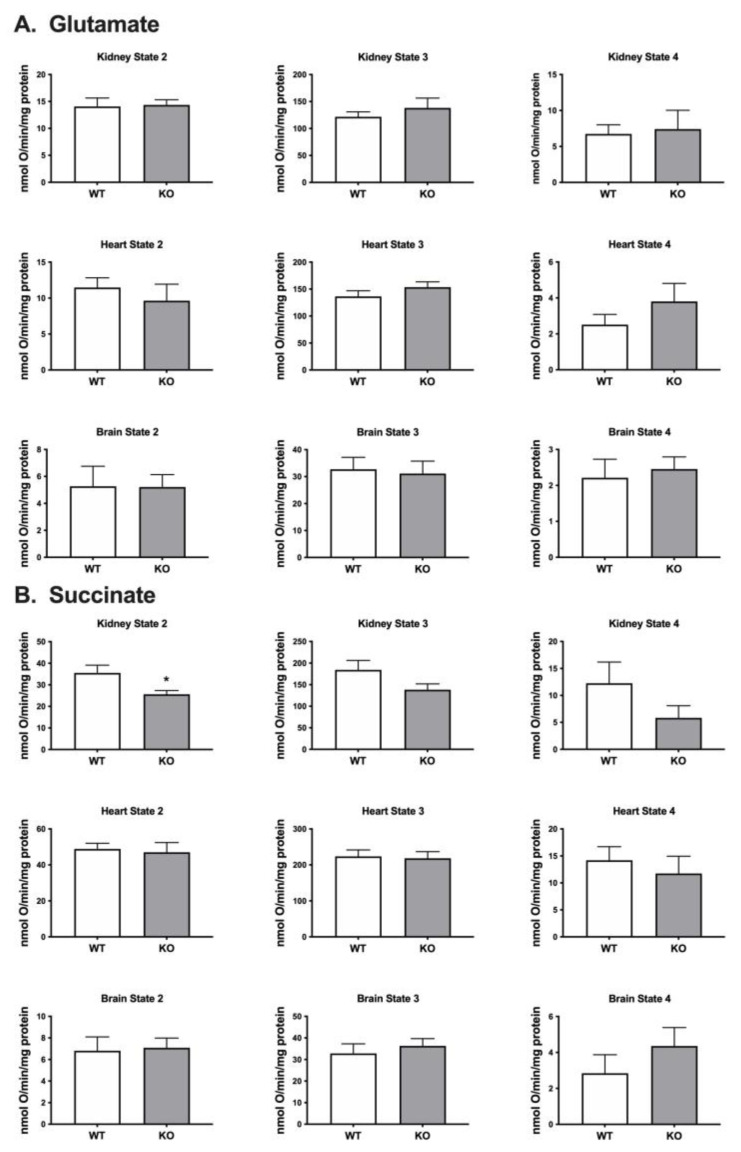
*Immp2l^KD^*−/− KO impact on respiration in mitochondria isolated from mouse organs. Respiration in mitochondria isolated from organs of wild-type mice and *Immp2l^KD^*−/− KO mice using glutamate (**A**) and succinate (**B**) as respiratory substrates. Respiration was measured under State 2 conditions (substrate only), State 3 (ADP present) or State 4 (chemical inhibition of ATP synthesis by oligomycin). Data represent mean ± SE (*n* = 6–8) and were analysed using an unpaired *t*-test * *p* < 0.05.

**Figure 4 ijms-25-00990-f004:**
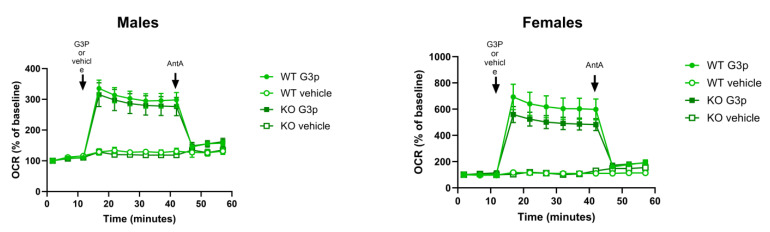
*Immp2l^KD^*−/− KO lowers G3P substrate mediated mitochondrial respiration in brown adipose tissue (BAT). Oxygen consumption rate (OCR) of frozen brown adipose tissue from wild-type (n = 3) and *Immp2l^KD^*−/− KO (n = 3) male and female mice. Glycerol 3-phosphate (G3P, 5 mM) and Antimycin A (AntA, 4 μM) were injected at the indicated times. OCR rates are represented as percentage of baseline. n = 3 wells per sample; values are represented as mean ± SE.

**Figure 5 ijms-25-00990-f005:**
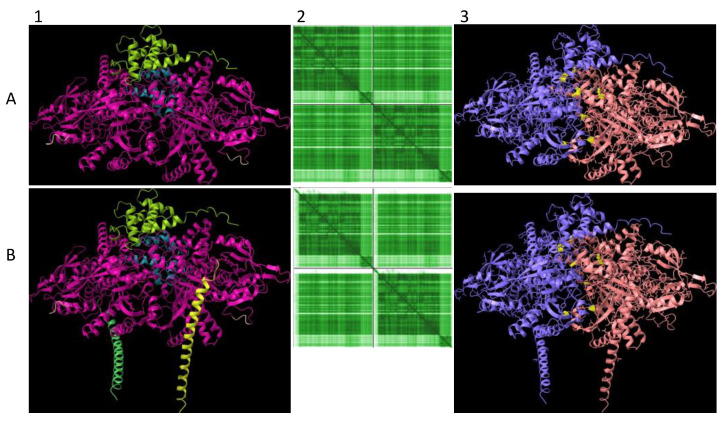
Mouse Gpd2 dehydrogenase homodimer structural variants. AlphaFold2-Multimer high-accuracy prediction of mouse Gpd2 enzyme homodimeric structures [13,14,15] with predicted alignment error (PAE) plots visualised through ChimeraX. Rows: (**A**) ‘cleaved Gpd2′ enzyme homodimer and (**B**) ‘uncleaved Gpd2′ enzyme homodimer found in the *Immp2l^KD^*−/− KO mouse. Columns: (**1**) Dimer predicted alignment error (PAE) colour comparison. (**2**) PAE plots where green colour intensity depicts higher confidence. Upper left and lower right quadrants represent the confidence of the predicted structure of the monomeric enzyme whereas lower left and upper right quadrants depict the confidence of the predicted dimerization of the enzyme subunits. (**3**) Dimer (Chain A, blue; Chain B, wood colour) inter-subunit connections (yellow) with 156 inter-subunit connections predicted for the ‘cleaved Gpd2′ dehydrogenase homodimer in mouse compared with 166 connections predicted for the ‘uncleaved Gpd2′ dehydrogenase homodimer.

**Figure 6 ijms-25-00990-f006:**
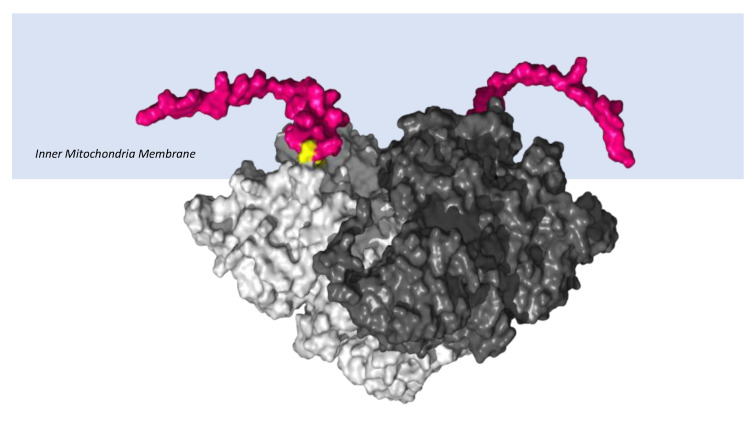
Surface structure of ‘uncleaved Gpd2′ dehydrogenase homodimer. AlphaFold2-Multimer [13,14,15] predicted surface structure of the ‘uncleaved Gpd2′ enzyme homodimer embedded within the inner mitochondrial membrane of the *Immp2l^KD^*−/− KO mouse. The two subunits of the Gpd2 enzyme homodimer (different shades of grey), uncleaved N-terminal signal transit peptides (pink) with visible peptide cleavage recognition sites (yellow). Note: Mouse Gpd2 shares 30% homology with *E coli* GlpD, upon which this structural rendition was modelled [2].

**Figure 7 ijms-25-00990-f007:**
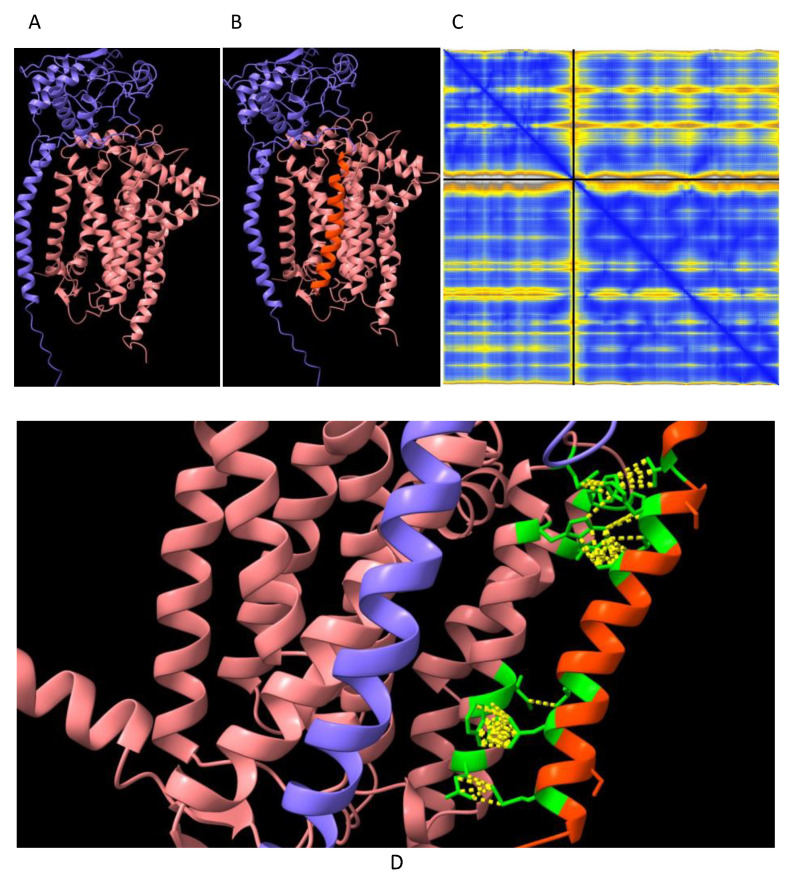
Cyc1 variant connections with the Cyb respiratory subunit of Complex III. AlphaFold2-Multimer high-accuracy prediction of mouse Cyc1-Cyb of Complex III [13,14,15] with predicted alignment error (PAE) plots visualised through ChimeraX. (**A**) ‘cleaved Cyc1’ interaction with Cyb. (**B**) ‘uncleaved Cyc1’ interaction with Cyb as found in *Immp2l^KD^*−/− KO mouse. (**C**) PAE plot of ‘uncleaved Cyc1’-Cyb where blue colour intensity depicts higher confidence. Upper left and lower right quadrants of PAE plot represent the confidence of the predicted structure of the unitary respiratory subunits whereas the lower left and upper right quadrants depict the confidence of the predicted respiratory subunit complex of Cyc1 (blue), Cyb (wood) and the ‘uncleaved’ signal transit peptide of Cyc1 (orange). (**D**) Magnified lateral view of subfigure B (above), highlighting the contact forming residues between the ‘uncleaved’ signal transit peptide (Orange) of Cyc1 (blue) and Cyb (wood). The contact forming residues are shown in green and the contacts are shown as yellow dotted lines.

**Figure 8 ijms-25-00990-f008:**
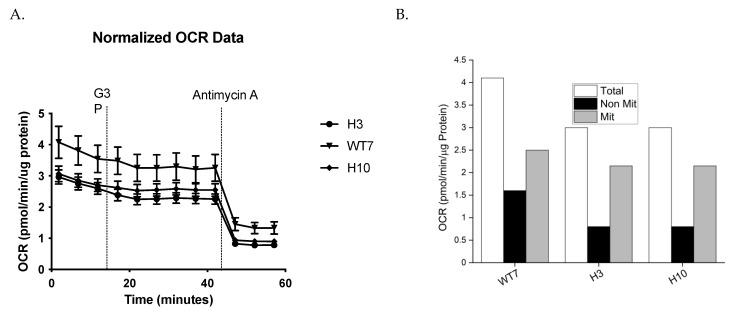
*Immp2l^KD^*−/− KO lowered oxygen consumption rates (OCR) in primary mouse embryonic fibroblast cell lines (MEFs). (**A**) OCR Time Course. (**B**) Mean OCR. Primary MEFs generated from embryo litter mates were harvested at the same time & grown & analysed under identical conditions in DMEM cell culture media for the determination of OCR. Two MEFs derived from homozygote male *Immp2l^KD^*−/− KO mouse embryos ‘H3’ and ‘H10’ were compared with the OCR of one male wildtype litter mate MEF ‘WT7’.

**Table 1 ijms-25-00990-t001:** Comparative analysis of respiration rates in mouse embryonic fibroblast cell lines (MEFs).

MEF Cell Line	TOTAL OCR pmol/min/µg Protein	Non-Mitochondrial Respiration	Mitochondrial Respiration
Wildtype—WT7	4.1	1.6 (39% of Total)	2.5 (61% of Total)
Immp2l KO—H3	3.0 (73%)	0.8 (50%)	2.2 (88%)
Immp2l KO—H10	3.0 (73%)	0.8 (50%)	2.2 (88%)

## Data Availability

Data are contained within the article.

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
