# Peer review of "Immp2l Enhances the Structure and Function of Mitochondrial Gpd2 Dehydrogenase"

_ijms, 2024, doi:10.3390/ijms25020990_

Round 1
Reviewer 1 Report
Comments and Suggestions for Authors
The main question addressed by the research is the role of IMMP2L peptidase in enhancing mitochondrial function, specifically its interaction with GPD2 dehydrogenase and its impact on cellular respiration. The study aimed to investigate the effect of IMMP2L on whole cell respiration by generating and testing primary fibroblast cell lines derived from IMMP2L knockout mouse embryos. The topic is both original and relevant in the field of mitochondrial biology. The study addresses a specific gap in the field by providing new insights into the role of IMMP2L peptidase in enhancing the structure and function of mitochondrial GPD2 dehydrogenase, which has not been previously explored in depth. The findings have significant implications for understanding mitochondrial function and potential therapeutic targets, particularly in the context of disorders related to mitochondrial dysfunction.
This research adds to the subject area by providing novel insights into the specific role of IMMP2L peptidase in enhancing the structure and function of mitochondrial GPD2 dehydrogenase. The study's findings contribute to a deeper understanding of mitochondrial dynamics, cellular respiration, glycolysis, metabolism, and the production of mitochondrial reactive oxygen species (mitoROS) . Additionally, the study highlights the potential impact of IMMP2L on NAD+ metabolism, gene expression, and mitochondrial dynamics, providing a comprehensive view of its effects on cellular homeostasis. These contributions distinguish this research from other published material and expand our knowledge of the intricate mechanisms underlying mitochondrial function.
The conclusions of the study are consistent with the evidence and arguments presented, and they address the main question posed. The study provides compelling evidence that IMMP2L peptidase plays a crucial role in enhancing the structure and function of mitochondrial GPD2 dehydrogenase, which in turn optimizes and enhances mitochondrial respiration. The study's findings are supported by a range of experiments, including analyses of primary fibroblast cell lines derived from IMMP2L knockout mouse embryos, body composition analyses, mitochondrial and gene expression analyses, and more. The authors' conclusions are well-supported by the data presented and provide valuable insights into the role of IMMP2L in mitochondrial function and cellular homeostasis.
Considering the issues presented above, it is appropriate to publish the study in its current form.
Reviewer 2 Report
Comments and Suggestions for Authors
The article explores the role of the 'Inner Mitochondrial Membrane Peptidase 2 Like' (IMMP2L) in regulating the structure and function of mitochondrial glycerol phosphate dehydrogenase 2 (GPD2). The study utilizes an IMMP2L knockout (Immp2lKD-/- KO) mouse model to investigate the impact of IMMP2L deficiency on organ size, mitochondrial respiration, and non-mitochondrial respiration. While the study provides interesting insights into the potential role of IMMP2L in cellular homeostasis, there are several aspects that warrant critical examination.
The article's structure is clear, with an introduction, results, and discussion sections. However, some sections are overly detailed, making it challenging for non-experts to grasp the key points.
The article references prior studies on IMMP2L but fails to provide a comprehensive comparison with existing literature. A critical review should evaluate how these new findings align or conflict with previous research, allowing readers to contextualize the significance of the current study.
The article reports various physiological and biochemical changes in Immp2lKD-/- KO mice. While the results suggest alterations in mitochondrial and non-mitochondrial respiration, the underlying molecular mechanisms remain unclear. A more in-depth discussion of the biological relevance and potential downstream effects of these changes is needed.
The article touches on potential implications for organ size, mitochondrial respiration, and non-mitochondrial respiration in the context of IMMP2L deficiency. However, the clinical relevance and translational aspects of these findings are not extensively discussed. A critical review should address the broader implications of the research for understanding diseases or potential therapeutic interventions.
The conclusion lacks a clear summary of the key findings and their significance. Additionally, the article could benefit from proposing future research directions to build upon the current study and address any remaining questions.
In conclusion, while the article provides valuable insights into the role of IMMP2L in mitochondrial function, a more critical examination is needed to enhance clarity, address experimental design considerations, and discuss the broader implications of the findings in the context of existing research.
Round 2
Reviewer 2 Report
Comments and Suggestions for Authors
The current version of the manuscript is suitable for publication in the IJMS.